# Stealing User Prompts from Mixture-of-Experts Models

## Abstract

Mixture of Expert (MoE) models improve the efficiency and scalability of dense language models by *routing* each token to a small number of experts in each layer of the model. In this paper, we show how an adversary that can arrange for their queries to appear in the same batch of examples as a victim's queries can exploit expert-choice routing to the full disclosure of a victim's prompt. We successfully demonstrate the effectiveness of this attack on a two-layered Mixtral model. Our results show that we can extract the entire prompt using $\mathcal{O}(\text{Vocabulary size} \times \text{prompt length}^2)$ queries or a maximum of 100 queries per token in the setting we consider. Our work is the first of its kind data reconstruction attack that originates from in a flaw in the model architecture, as opposed to the model parameterization.

## 1 Introduction

In recent years, the Mixture of Experts (MoE) architecture has emerged as a powerful and efficient approach for inference of large-scale machine learning models (Jiang et al., 2024; Shazeer et al., 2017; Aljundi et al., 2017; Eigen et al., 2013; Jordan and Jacobs, 1994; Jacobs et al., 1991), particularly in natural language processing (Shazeer et al., 2017; Fedus et al., 2022; Riquelme et al., 2021; Du et al., 2022). This architecture distributes computational load across multiple expert modules, each specializing in different aspects of the underlying task. In the context of language, this means dividing the input text into smaller units called 'tokens' and assigning them to the expert(s) best suited to handle them.

However, this specialization can also introduce new vulnerabilities. Prior work from Hayes et al. (2024) identified a critical architectural flaw in MoE models prone to *token dropping*. Token dropping refers to a phenomena that occurs when an expert, already at full capacity, gets assigned additional tokens; these additional tokens cannot be processed and, as a result, are either routed to other experts or dropped. Hayes et al. demonstrated that if an adversary can ensure their data is processed in the same batch as another user's data, they can exploit token dropping to launch denial-of-service attacks, by filling the buffers of experts that the user's data relies on. The attack of Hayes et al. reduced quality of the model responses to victim queries, causing denial of service.

Building on this foundation, we show that the same vulnerability has far more significant consequences and can be exploited to compromise user privacy. We demonstrate that by carefully crafting a batch of inputs, an attacker can manipulate the expert buffers within the MoE model, leading to the full disclosure of a victim's prompt that is included in the same batch. This represents the first data reconstruction attack of its kind that originates in a flaw within the design of the model's architecture, rather than in its learned parameters. The core principle underpinning this attack is that token dropping introduces a shared information channel; if one user's data can affect the routing pattern of another user's data, this introduces an information channel (albeit a low entropy one) that can be used to infer which tokens were in another user's message. In other words, we show that the MoE architecture that uses Expert Choice Routing (Zhou et al., 2024) is vulnerable to **Conditional Adversarial Token-Dropping attack**, where an attacker can infer what data a victim submitted to the model. A high level description of Conditional Adversarial Token-Dropping attack is given in Figure 1. Although our findings are limited to a specific choice of token to expert routing algorithm (Zhou et al., 2024), we hypothesise that other MoE routing strategies that break the fundamental assumption of independence across a batch of inputs may also be vulnerable.

In this paper we make the following contributions:

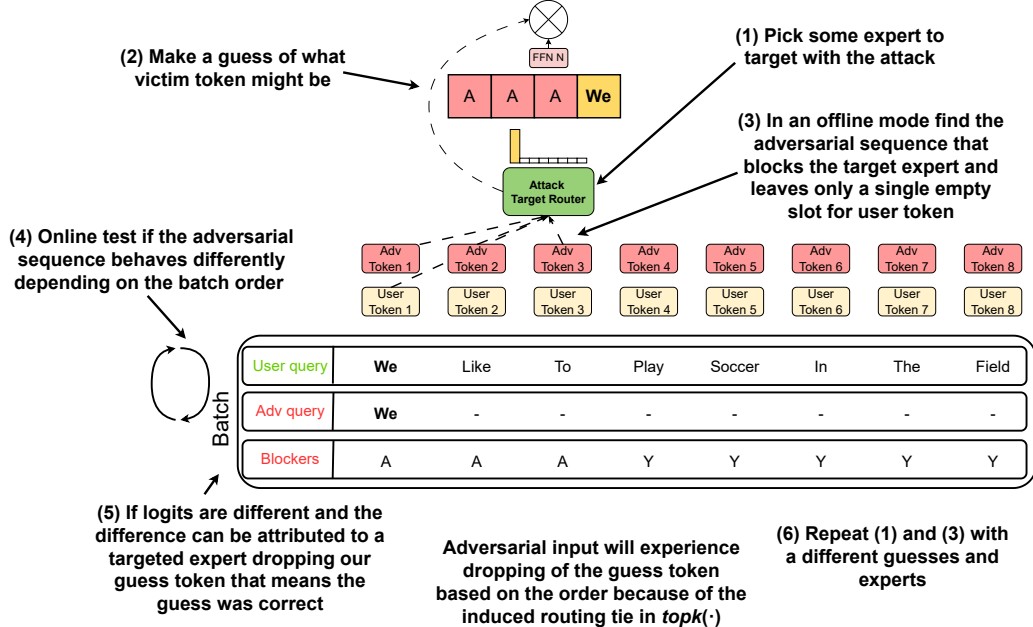

Figure 1: We show the overall flow of Conditional Adversarial Token-Dropping, following the example Expert Choice Routing from Zhou et al. (2024) depicted in Figure 7. The adversary finds a set of tokens that fill the buffer of a given expert such that only a single space is left empty. In the example above, the adversary fills the expert with 'A', such that the only spot that is left is for token 'We'. Now, the adversary makes a guess of what that token may be and observes what happens to the logits. If the logits do not change when you process the batch in a different order, that means that tie-handling has triggered for the same tokens. This allows the adversary to deduce that the guess was correct. If it was incorrect, another guess is made and tested. Note that only two queries are needed to test a given guess.

- We demonstrate the first data stealing attack, Conditional Adversarial Token-Dropping, that originates in the MoE model architecture;

- We demonstrate feasibility of Conditional Adversarial Token-Dropping on a Mixtral backbone (Jiang et al., 2024) with the Expert Routing strategy; we show that an attacker can verify their guess for an unknown user message *in just two queries*, while a general extraction attack scales by $\mathcal{O}(2^{L \times n} \times |\mathcal{V}|)$ *local queries* to a model hosted by the attacker and by $\mathcal{O}(|\mathcal{V}| \times l^2)$ *remote queries* to a target model that is being attacked for $L$ layers, $n$ experts, and a sequence of length $l$ with a vocabulary of size $|\mathcal{V}|$;

- Finally, we discuss defences against Conditional Adversarial Token-Dropping.

## 2 RELATED WORK

### 2.1 MIXTURE OF EXPERTS

The concept of Mixture of Experts (MoE) was first introduced by Jacobs et al. (1991) and Jordan and Jacobs (1994), but has more recently become a popular tool for efficient inference on transformer based LLMs (Shazeer et al., 2017; Zoph et al., 2022; Renggli et al., 2022; Jiang et al., 2024), primarily because it allows the model to selectively activate only a small fraction of the total parameters for any given input. An MoE layer in a large language model (LLM) consists of $n$ expert modules and a gating function, which routes a token to an expert (or subset of the $n$ experts). Each expert module can specialize in different aspects of language tasks (e.g. coding) and the gating function decides which experts are relevant for a particular input. Since only a subset of experts is activated for an input, the number of parameters activated is significantly smaller than the overall number of

parameters in the LLM, which translates to fewer floating-point operations and faster inference. This in turn allows one to build extremely large networks without a corresponding increase in inference costs. Some of the best performing modern LLMs utilize MoE architectures e.g. Gemini-1.5 (Team, 2024) , Mixtral (Jiang et al., 2024), and Grok-1 (xAI, 2023).

## 2.2 VIOLATIONS OF USER PRIVACY

Although previous work has investigated how user privacy can be compromised in LLMs (Debenedetti et al., 2023; Shen et al., 2024), none have thus far investigated vulnerability of user data privacy due to the underlying model architecture, specifically how input representations can be influenced by other data within the same processing batch. Hayes et al. (2024) demonstrated previously that batch composition can be used by adversaries to exploit MoE routing and to launch denial-of-service attacks; we instead exploit it to leak private user supplied prompts. Note that Expert Choice Routing considered in this work is one of many different routing strategies that breaks the implicit batch independence; we hypothesise that other routing strategies may be similarly vulnerable.

## 3 BACKGROUND

### 3.1 THREAT MODEL

In this paper, we make the following *simplifying* assumptions. First, we assume that the adversary has *white-box access* to the model that uses an MoE with *cross-batch Expert Choice Routing* strategy (Zhou et al., 2024). This can apply in a setting where a third party is using the base model that is available publicly e.g. implementation available through t5x (Roberts et al., 2022). Second, the adversary can *control the placement of its and the user inputs in the batch*. Third, the adversary can query the model repeatedly ensuring that the the user supplied input is consistently in the same batch as its own inputs; the adversary and user inputs are always batched together and sent to the model for processing. At face value, this threat model is *unrealistic* as the second and the third assumptions require access to the serving infrastructure to control in-batch placement. However, the ability to mount man-in-the-middle attacks over networks is a commonplace assumption in security research. We defer a more detailed discussion of the practicalities of the attack and potential methodological improvements to Section 6.

### 3.2 PRIMER ON LANGUAGE MODELS AND MIXTURE OF EXPERTS

A transformer based large language model is a function $f_\theta : \mathcal{V}^l \to \mathcal{P}(V)$ that takes as input a sequence of *tokens* from a vocabulary $\mathcal{V}$ and outputs a probability distribution over the vocabulary, $\mathcal{P}(V)$. In particular, we are interested in functions of the form $f_\theta(z) = \mathrm{softmax}(W \cdot h_\theta(z))$, where $W$ is an unembedding matrix and $W \cdot h_\theta(z)$ gives a set of *logits* over $\mathcal{V}$.

We assume that the model $h_\theta$ consists of multiple MoE layers. A MoE layer consists of $n$ expert functions $\{e_1, e_2, \ldots, e_n\}$ where $e_i : \mathbb{R}^d \to \mathbb{R}^d$ is a feed forward layer that takes in $d$-dimensional token representations and outputs new features of the same dimensionality. The MoE layer also consists of a gating function $g : \mathbb{R}^d \to \mathbb{R}^n$ which is used to assign token representations to experts by outputting a probability distribution over the $n$ experts.

Large language models are commonly ran on *batches of inputs* to improve hardware utilization and efficiency. This means $f_\theta$ in reality operates on the domain $\mathcal{V}^{l \times b}$, where $l$ is the sequence length of an input, and $b$ is the batch size. For models that do not use MoE layers, the computation is entirely parallel over a batch of inputs; the computations of one input in the batch cannot affect the computations of another input in the batch. For models that do use MoE layers, this is no longer true, as the gating function $g$ can only assign a limited number of token representations from a batch to a specific expert. There are many different choices for how to assign tokens to experts given the output of the gating function; this is also commonly known as the *routing strategy* (Cai et al., 2024).

In this work, we focus on Expert Choice Routing which allows each expert to independently select its `topk` assigned tokens from a batch of tokens (Zhou et al., 2024). The value $k$ represents the fixed capacity of each expert, signifying the number of tokens it can process; we refer to this as

the *expert's buffer capacity*. This inherently ensures a balanced load across experts and introduces flexibility in allocating computational resources. In our experimental setup, we define $k$ as:

$$k = \frac{t \times c}{n} \tag{1}$$

Here, $t$ represents the total number of tokens in the input batch (typically batch size $b$ multiplied by sequence length $l$), $c > 0$ is the capacity factor, indicating the average number of experts each token utilizes, and $n$ is the total number of experts. Let $Z \in \mathbb{R}^{t \times d}$ denote a batch of input token representations at a given layer, where $d$ is the hidden dimension of the model. For each $z_i \in Z$, we compute $g(z_i) = \{p_{i1}, p_{i2}, \ldots, p_{in}\}$, which outputs a probability distribution over the $n$ experts. This produces the matrix:

$$G = \begin{bmatrix} p_{11} & p_{12} & \cdots & p_{1n} \\ p_{21} & p_{22} & \cdots & p_{2n} \\ \vdots & \vdots & \vdots & \vdots \\ p_{t1} & p_{t2} & \cdots & p_{tn} \end{bmatrix} \tag{2}$$

where $p_{ij}$ represents the probability of assigning token $z_i$ to expert $e_j$. Expert Choice Routing applies a column wise top$k$ selection of tokens; token $z_i$ is routed to expert $e_j$ if $p_{ij}$ is one of the top$k$ probabilities in column $j$. Unlike other routing strategies, where experts may handle a variable number of tokens (Fedus et al., 2022; Lepikhin et al., 2020; Shazeer et al., 2017), in Expert Choice Routing the expert load is perfectly balanced by design, each expert handles exactly $k$ tokens.

Observe that not all tokens within the batch may be processed by an expert. For example, if $c$ is small (e.g. $<< 1$) then the number of tokens processed by each expert is substantially smaller than $t$, the total number of tokens in the batch. In such cases, tokens that are not assigned to any expert are dropped – that is, not processed by any expert (Fedus et al., 2022; Hwang et al., 2023). This is commonly assumed to be of little consequence, as it is standard for MoE models to have residual connections between layers, meaning that the effect of dropping a token is limited. However, we will show that token dropping can introduce a shared information side channel which can be exploited.

## 4 CONDITIONAL ADVERSARIAL TOKEN-DROPPING ATTACK

This section details our two exploits against models that use MoE architectures with the Expert Choice Routing strategy. Both exploits leverage the primitives explained in Section 4.1 to compromise user message privacy. The attacks are:

1. **Oracle attack**: The attacker guesses the victim's prompt and verifies it with only two queries to the model. (Described in detail below)

2. **Leakage attack**: The attacker sequentially executes the oracle attack to extract the victim's prompt without prior knowledge.

We successfully demonstrate the leakage attack on a **two-layered** Mixtral model (Jiang et al., 2024) with Expert Choice Routing strategy (results are in Section 5, by two-layer we mean we only use the first two layers of a larger model).

At a high level, the attacks are based on the ability to create dedicated adversarial inputs (Figure 2) that when included in a batch, intentionally shape what is included in an expert's buffer in the first MoE layer (Figure 3) in such a way that information about user prompts within the batch leaks. In particular, the adversary finds a set of tokens, which we refer to as *blocking tokens*, that have high priority of being routed to an expert. The adversary then guesses the target token that they are trying to leak from the victim input, and uses the blocking tokens to ensure that this (guessed) token is at the boundary of the expert buffer (i.e. has the $k^{\text{th}}$ priority). The reader will recall from Section 3.1 that the adversary can control the position of the victim's message within the batch, and so creates an adversarial batch that consists of blocking tokens, the guessed message and the victim's message (the message they are trying to infer). The adversary then sends two queries: one where the adversarial batch contains the victim's input in a batch position before the guessed input token, and one where the the guessed input token is before the victim's input. If the victim and

guessed inputs are identical, then the model output on these two queries will be different, as either the victim or guessed message target token will be dropped depending on their order within the batch. Importantly, the guessed message tokens that are processed will be different in each query (in one query the guessed token is dropped and in another it is not), which causes a difference in model outputs. By equating difference in outputs to token dropping, the adversary can infer if their guess was correct.

We concentrate on the description of the oracle attack, as the token-by-token leakage attack is simply a sequential execution of the oracle attack. Next, we give a more detailed overview of the attack along with the primitives we make use of for execution.

## 4.1 ATTACK OVERVIEW

The goal of the attack is to force a scenario in which an attacker's token is conditionally dropped depending on the relative order in the same batch of an identical token belonging to the victim. We discuss the attack in more detail in Appendix E, but at a high level the exploit is triggered by the following steps:

1. **Step 1: Guess the next Token and Position:** The attacker guesses the target token and its position in a chosen expert's buffer, assuming the prefix is known (initially empty).

2. **Step 2: Construct the Adversarial Batch:** Using the primitives described in Section 4.2, the attacker crafts an adversarial batch that:
   (a) Places *blocking tokens* to fill the expert buffer, leaving one spot for the guessed token.
   (b) Includes the *probe sequence* with the known prefix and guessed token, with the goal of triggering tie-handling.
   (c) Adds a padding sequence to extend the buffer size and ensure stable sorting.

3. **Step 3: Send Two Queries:** The attacker sends the adversarial batch twice, changing the order of the victim's message and the *probe sequence*.

4. **Step 4: Map Observed Logits to Routing Paths:** The attacker uses a local model to find a mapping between observed logits to *routing paths* of the *probe sequence*.

5. **Step 5: Verify the Guess:** A correct guess is indicated if the guessed target token is routed to the chosen expert in the first query (where the *probe sequence* comes first) but not in the second query (where the victim's message comes first). This difference in routing paths arises because the input order of identical tokens influences the tie-breaking behavior. If the guess is incorrect, the routing paths should be consistent regardless of the input order.

## 4.2 ATTACK PRIMITIVES

We now describe the primitives behind each of the attack steps.

### 4.2.1 HANDLING TOKEN PRIORITY TIES IN EXPERT ROUTING

In Expert Choice Routing, each expert buffer serves as a priority queue. As discussed in Section 3.2, priorities are given to each token by way of the matrix $G$ of shape $(t, n)$, where $t$ is the total number of tokens in the batch and $n$ is the total number of experts. With token dropping the `topk` most prioritized tokens in each column of $G$ will be processed by the corresponding expert and the rest will be dropped. If two tokens have identical priority (they have the same probability of assignment to an expert) and they have priority $k$, then which of the two tokens is selected and routed to the expert is entirely down to the tie-handling behavior of the `topk` algorithm implementation. This is a feature (further discussed in Appendix B) in the CUDA implementation of `topk` which is based on a stable sort, with a "first come first serve" tie breaking policy. This distinguishable behavior which is a function of the relative order to tokens within a batch is *what we exploit in Conditional Adversarial Token-Dropping*.

## 4.3 EXTENDING AN EXPERT'S BUFFER SIZE

Extending the expert's buffer size, $k$, is crucial to the attack for two reasons:

- To make sure the victim's token are not dropped by default.

- The expert buffer capacity needs to be $> 32$ for predictable behavior in CUDA `topk` tie breaking (see Appendix B).

The expert buffer capacity is given in Equation (1).

In essence, it is the expected number of tokens per expert factored by the *capacity factor* $c$, which usually sits in the range of $[0.6, 2]$ (Gale et al., 2023). By sending a long query the total number of tokens will increase, the other inputs will be padded to match the sequence length $l$ and thus the effective size (or space) in each expert buffer will increase, as padding tokens are masked to achieve the lowest priority possible.

### 4.4 CONTROLLING TARGET-TOKEN PLACEMENT WITH BLOCKER SEQUENCES

Each token is assigned a priority (distribution over experts) for each expert; tokens with lower priority are pushed towards the end of the expert buffer, or dropped altogether if their priority is larger than $k$. We leverage local access to the model weights to pre-compute a sequence per expert with tokens that are assigned high-priority in the first MoE layer. We refer to this as a *blocking sequence*, as these tokens block other tokens from being assigned to an expert because of a higher priority. Since our goal is to place the target token at the edge of the expert buffer we need to guess its position in an expert buffer. We will then use our pre-computed blocker sequences to precisely set the correct number of blocker tokens needed within the adversarial batch. We further discuss the details of the approach we took for finding blockers in Appendix C.

Notice that although target token can be positioned in any position in an expert's buffer, the attacker controls all of the tokens in the batch except for the victim's. Therefore the position search-space is bounded to the victim's message length.

Figure 2: The adversarial batch consists of the four components: secret message of the victim; known prefix to the adversary that includes a new adversarial guess ; blockers that are used to shape the buffer of the target expert; and a long sequence that is used to extend the expert capacity, and also make `topk` predictable.

### 4.5 RECOVERING TARGET TOKEN ROUTING PATH

In MoE-based models, each token can be processed by at most $2^{n \times L}$ different expert combinations, where $L$ is the number of model layers. For example if a model contains 2 MoE layers and each layer contains $n = 8$ experts, then a token could be routed in $2^{16}$ unique ways through the model. If it is possible to map a model output to the routing path of the token, then we can detect whether token-dropping in the first MoE layer took place. We discuss the approach we took in further detail in Appendix D.

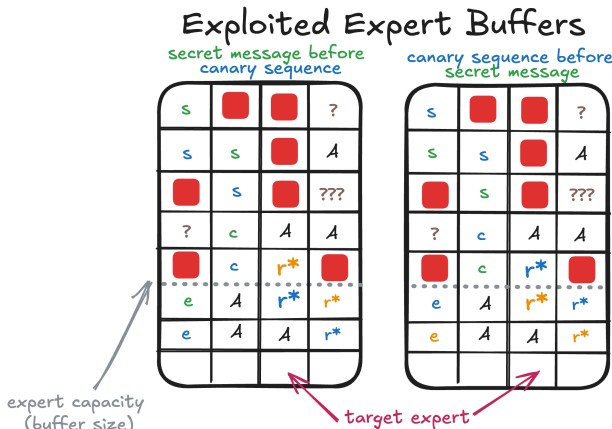

Figure 3: The buffer of the experts chosen to be exploited is filled with specially chosen blocking tokens that leave only a single position for the guessed target token. If the guessed token is correct, model outputs for the adversarial batch should be identical even if we permute the order of inputs. This is because the token that ends up in the buffer is the same if and only if tie-handing between tokens has been triggered and the guessed token has priority identical to victim's target token.

### 4.5.1 Do we really need token routing path maps?

Initially, one may question if computing token routing paths is necessary. We now discuss why this is crucial to the attack for models with more than one MoE layer.

In a single layer transformer model that utilizes MoE, the layers of the model are: (1) embedding, (2) attention, (3) MoE, (4) unembedding. Importantly, the attention layer does not affect the representation of the target token after token collisions due to priority ties in the MoE layer. A collision would have an immediate impact on the model output (logits). The attack is performed sequentially, token-by-token, the attacker needs to correctly guess the target token, and its position in a chosen expert. Any change to the logit output will then indicate this token was affected by token dropping.

For a two layer model, the layers are: (1) embedding, (2) attention, (3) MoE, (4) attention, (5) MoE, (6) unembedding. The second attention layer *does* affect the representation of the target token after the token collisions due to priority ties in the MoE layer. We therefore present an attack that handles the effect of attention that may interfere with our ability to understand how and when tokens are dropped. The second layer of attention is impacted by MoE token dropping, and that means that the prefix could also be dropped due to the impact of the adversarial batch and the unknown suffix of the victim message. Prefix tokens will also change the representation of the target token and thus its routing path through the second MoE layer, and therefore a method to recover the target token routing path given the model output is necessary.

## 5 Evaluation

**Setting** We evaluate our attack on the **first two transformer blocks** of `Mixtral-8x7B` (Jiang et al., 2024), using PyTorch 2.2.0+cu118. We then change the model to use the Expert Choice Router as is described by Zhou et al. (2024). This ensures that experts behave in an realistic manner, although used in a setting that they were not originally developed for. We restrict the vocab for guesses to lowercase letters with space of size 27, and limit our extraction messages to [`hello`, `world`, `admin`, `words`, `message`, `leakage`, `attacks`, `prompts`, `top secret`, `a password`, `vulnerable`, `token leak`, `side channel`, `exploitation`, `confidential`, `an adversary`, `buffer overflow`, `privacy attacks`, `ssn aaa bb cccc`, `secret messages`]. We use a restricted vocab to only 9,218 out of 32,000 tokens for finding blockers, which we discuss in detail in Appendix C. Finally, we quantise the router weights to 5 digits to induce ties.

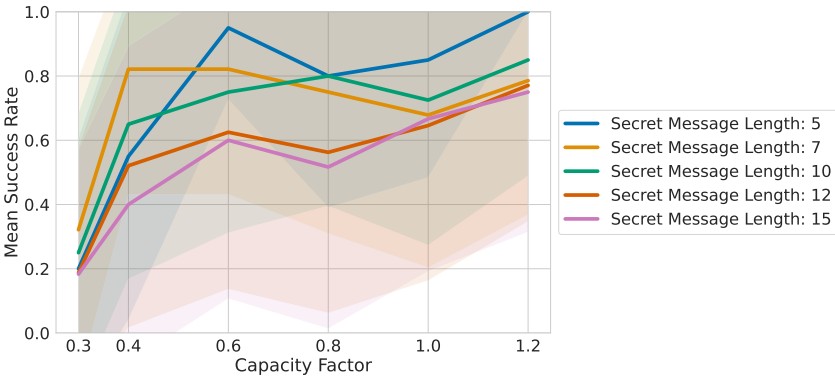

Figure 4: Extraction performance of user queries for messages of different sizes. Individual scores are presented in Figure 10–Figure 14. Performance of the Conditional Adversarial Token-Dropping improves together with the capacity factor, at the same time extracting longer messages is harder.

.

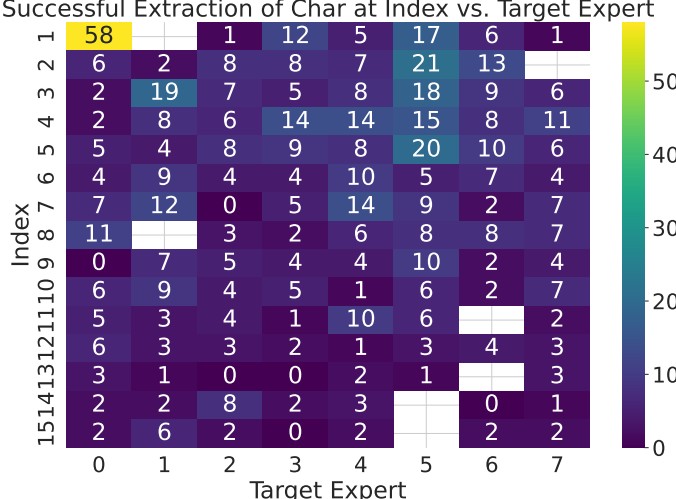

Figure 5: Heatmap showing the correlation between the expert and the index of the input token where the attack succeeds. Here, the attack progresses to the next token when any expert is successfully exploited to leak the token of the victim.

**Conditional Adversarial Token-Dropping** We find that it is possible to extract secret user data for all of the possible inputs we considered. As illustrated in Figure 4, the success rate of data extraction varies with both the expert capacity factor and the message size. Specifically, increasing the capacity factor improves the ability to extract victim tokens. Conversely, longer secret messages exhibit higher error rates, likely due to the increased complexity of shaping expert buffers and the heightened potential for cross-interference and unintended token drops in the later stages of extraction.

**What experts leak the victim tokens?** Figure 5 shows the expert and the index where the attack succeeds. Note that Conditional Adversarial Token-Dropping moves to the next token when a successful attack is found on any of the experts, meaning that the plot shows the index and the target expert that tend to work first for extraction. In theory, we could have used any expert, yet in practice we find that expert one and five are often exploited to leak the victim tokens. Figure 8 and Figure 9 show the relationship between the expert–position and the expert–token respectively for successful victim token leakage.

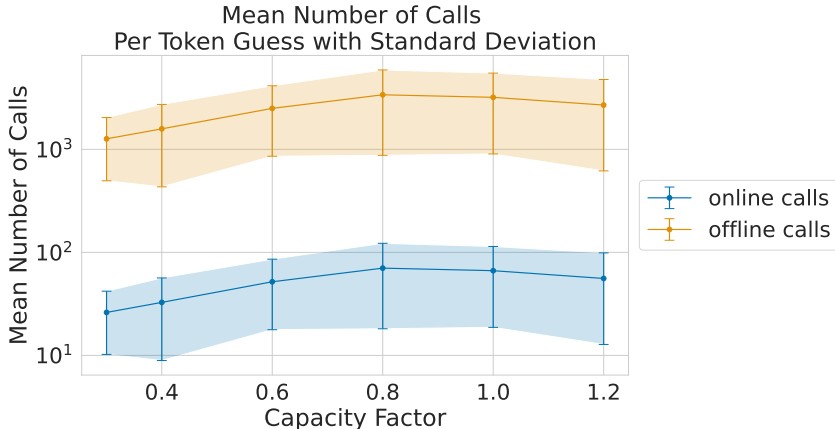

Figure 6: Number of queries required to leak a token of the victim. The majority of queries are performed locally by the adversary on the local (offline) copy of the model, with only a fraction of online queries (to target model) required to execute the attack. Across all capacity factors, leaking a single token requires up to 100 online queries.

**What is the cost of leaking the tokens?** Figure 6 shows the required number of queries to leak a given token. We find that majority of the cost is carried by the adversary locally on their own copy of the model, while online calls to the model with victim data present are not as significant. We find that for all of the capacity factors per token leakage requires up to a hundred queries to the target (online) model with victim data.

## 6 DISCUSSION

**Methodological improvements** In this paper we show that it is possible to exploit Expert Choice Routing to extract private victim data placed in the same batch as an adversary's data. Conditional Adversarial Token-Dropping currently requires 2 queries per guess for verification, and $2^{n \times L}$ per token queries for general extraction. This makes it infeasible at present to use our attack against real world systems. Yet, we believe that performance of the attack could be significantly improved. Firstly, we hypothesize that refining the buffer shaping process could enable the selection of blockers that prevent inter-batch token interference. We discuss this further in Appendix C. Secondly, we suspect that an alternative approach exists to determine the processed token without exhaustively constructing all possible $2^{n \times L}$ expert combinations, potentially by learning a mapping between outputs and routing paths. We discuss this further in Appendix D. Thirdly, targeting the final MoE layer instead of the first may eliminate the need for routing path tracking altogether. Fourth, the current attack requires precise matching in its exploitation for tie-handling. We hypothesise that relative placement of tokens can similarly be used to signal what victim token is used. Finally, we believe that a black-box variant of this attack is feasible – a local clone of the model at present is only used to find blocking sequences and for inverting token routing paths. We hypothesise that both of the tasks could inefficiently be deduced from black-box access.

**Optimisation–Security trade-off** Within the domain of computer security, it is well-established that prioritizing performance optimization often inadvertently introduces vulnerabilities to side-channel attacks (Anderson, 2010). In the case of MoE models, the Expert Choice Routing strategy, which optimizes for efficiency, inadvertently creates a side channel that allows the attacker to exploit the model. Our work highlights the importance of rigorous adversarial testing of any optimization introduced into machine learning pipelines to safeguard user privacy. While we focus on a specific routing strategy, we anticipate that similar vulnerabilities may exist in other strategies that violate implicit batch independence.

**Defences** Having established general vulnerability of MoE-based models with Expert Choice Routing, we now shift our focus to potential defense strategies. A crucial first step in mitigating these vulnerabilities is to preserve in-batch data independence, particularly across different users. This en-

sures that adversaries cannot exploit the routing strategy. Second, the current attack design requires precise shaping of the expert buffer, therefore, introducing any form of non-determinism into the system can effectively disrupt the attacker's ability to exploit this vulnerability. This could involve incorporating randomness into various aspects of the model, such as the expert capacity factor, the batch order, the input itself, or the routing strategy.

# 7 CONCLUSION

In classical dense LLMs, it is essentially impossible for one user's data to impact another user's output. But MoE models introduce a side-channel: one user's queries can impact a different user's outputs. The magntidue of this leak is minuscule challenging to detect. But by carefully crafting adversarial input batches, we show how to manipulate the expert buffers within the MoE model with Expert Choice Routing, leading to the full disclosure of a victim's prompt included in the same batch. At present, Conditional Adversarial Token-Dropping is only possible when Expert Choice Routing is used, yet we hypothesise that other routing strategies can be similarly vulnerable. While the current threat model assumes unrealistic attacker capabilities, we believe that future research can extend the practicality of these attacks.

More broadly, attacks such as this highlight the importance of system-level security analysis at all stages of model deployment, starting as the design of the architecture, and extending towards as late as the actual deployment of the model and how different user queries are batched together. Studying any one component in isolation may give the appearance of safety, but only when the system as a whole is analyzed is it possible to understand vulnerabilities such as this. We hope that future work will perform other analysis of this type on future advances.

# 8 REPRODUCIBILITY STATEMENT

To ensure reproducibility, we provide a comprehensive outline of our attack methodology in Section 4. We include all of the details about the attack and provide a detailed algorithmic description in Appendix E. Our evaluation in Section 5 relies on the base model that is openly available (Mixtral-8x7B). A number of supplementary figures in the appendix illustrate all of the details required to replicate the work.

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

## A    EXPERT CHOICE ROUTING

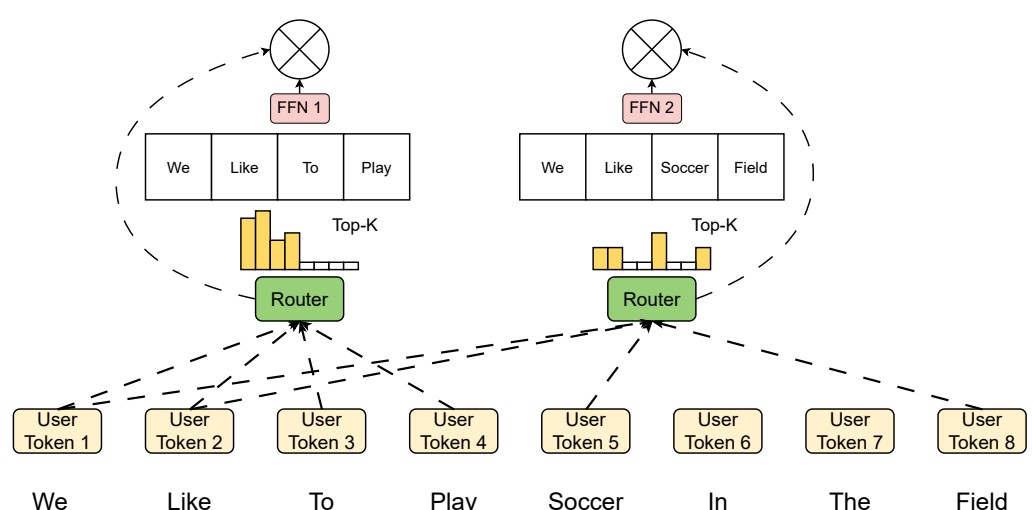

Figure 7: Figure is adopted from Zhou et al. (2024) and describes the vanilla Expert Choice Routing strategy. Here, tokens are routed to a pair of experts, where the experts choose `topk` tokens to process themselves. In the example above, for a `topk` with $k = 4$, the first four tokens are processed by the first expert, whereas the last 4 are processed by the second expert.

## B    EXPLOITING TIE-HANDLING IN TOPK IMPLEMENTATIONS

Our attack leverages consistent behaviour in tie-handling in `topk` sorting algorithms in a 2.2.0+cu118 PyTorch with CUDA environment. It is generally known in the community that the `topk` function is unstable on CPUs due to underlying parallelism (Issues, 2019; 2020; 2024). To best understand the impact of `topk` stability consider the code listing below. Here, we find that the returned indices from `topk` for CUDA tensors are in order (see outputs highlighted in green), while for CPU tensors it is not (see outputs highlighted in red). Suggesting that our method to verify token placement correctness in expert buffers is not going to work on CPUs.

```python
import torch
for device in ['cuda', 'cpu']:
  for size in [32, 33]:
    for is_sorted in [True, False]:
      print(size, is_sorted, device)
      print(torch.topk(
        torch.Tensor([1]*size).to(device),
          k=size, sorted = is_sorted).indices
      )
```

```
32 True cuda
tensor([31, 30, 28, 29, 25, 24, 26, 27, 19, 18, 16,
17, 21, 20, 22, 23,7, 6, 4, 5, 1, 0, 2, 3, 11, 10, 8, 9, 13, 12, 14, 15], device='cuda:0')
32 False cuda
tensor([ 0, 1, 2, 3, 4, 5, 6, 7, 8, 9, 10, 11, 12, 13, 14, 15, 16,
17, 18, 19, 20, 21, 22, 23, 24, 25, 26, 27, 28, 29, 30, 31], device='cuda:0')
33 True cuda
tensor([ 0, 1, 2, 3, 4, 5, 6, 7, 8, 9, 10, 11, 12, 13, 14, 15, 16,
17, 18, 19, 20, 21, 22, 23, 24, 25, 26, 27, 28, 29, 30, 31, 32], device='cuda:0')
33 False cuda
tensor([ 0, 1, 2, 3, 4, 5, 6, 7, 8, 9, 10, 11, 12, 13, 14, 15, 16,
17, 18, 19, 20, 21, 22, 23, 24, 25, 26, 27, 28, 29, 30, 31, 32], device='cuda:0')
```

```
32 True cpu
tensor([17,  0,  9, 10, 13, 14, 15, 12,  7,  6,  5,  4,  3,  2,  1,  8,
16, 18, 19, 20, 21, 22, 23, 24, 25, 26, 27, 28, 29, 30, 31, 11])
32 False cpu
tensor([16, 31, 30, 29, 28, 27, 26, 25, 24, 23, 22, 21, 20, 19, 18, 17,
8,  1,  2,  3,  4,  5,  6,  7, 12, 15, 14, 13, 10,  9,  0, 11])
33 True cpu
tensor([17,  0,  9, 10, 13, 14, 15, 12,  7,  6,  5,  4,  3,  2,  1,  8,
16, 18, 19, 20, 21, 22, 23, 24, 25, 26, 27, 28, 29, 30, 31, 32, 11])
33 False cpu
tensor([16, 32, 31, 30, 29, 28, 27, 26, 25, 24, 23, 22, 21, 20, 19,
18, 17,  8,  1,  2,  3,  4,  5,  6,  7, 12, 15, 14, 13, 10,  9,  0, 11])
```

## C  FIND BLOCKING SEQUENCES

To construct *adversarial batches* efficiently, the attacker has to be able to generate `blocking_sequences` tailored to specific attack parameters:

- $e_i$, Target expert;
- $p$, Priority / probability threshold;
- $nb$, Number of blockers.

Each expert is assigned a pre-computed sequence comprised of highly prioritized blocking tokens. This sequence can be truncated to accommodate attacks with lower priority targets or requiring fewer blocking tokens.

We search the highly prioritized blocking sequences **per expert**:

1. First we restrict the vocabulary to a set of prefix free tokens. With that we can freely append tokens without changing previous ones and avoid affecting the blocking;
2. We set a high **maximum** priority threshold and expect all $p$ to satisfy $p \leq P = 0.85$;
3. We bound $nb \leq NB = \frac{k-1}{b-3}$. As the total number of blockers needed in a batch is at most $k - 1$ and there are $b - 3$ blocking sequences in a batch;
4. We search for a sequence of length $SL \leq |\texttt{long\_sequence}|$ with $NB$ tokens satisfying $p_{e_i} < T$.

To find this blocker sequence we

1. Set `blocking_sequence` = init empty sequence with $< bos >$;
2. Randomly generate a candidate chunk of length $\frac{SL}{NB}$;
3. If it has at least one token satisfying $p_{e_i} \geq P$ we append it to `blocking_sequence`;
4. We trim unnecessary tokens at the end and repeat this loop until $N$ chunks are appended.

## D  RECOVERING THE TOKEN ROUTING PATH

To get a useful signal from the conditional token dropping we induced on the first MoE layer of the model we expect to observe a deviation in the model outputs of our *probe sequence*, depending on its position in the adversarial batch. However a change in model output can be attributed to a wide set of factors:

- Floating point errors;
- Dropping of prefix tokens due to unknown suffix tokens in the victim's message;
- Dropping of tokens in other MoE layers of the model;
- Double dropout in which two tokens are placed simultaneously at the edge of different experts buffers with an identical token.

It is therefore important to devise a method that can trace the deviations in model outputs to the expected token-dropping the attacker is aiming to induce. Our approach is robust but conservative; it is computationally slow and is the main bottleneck of our attack and the reason we didn't scale the attack to more than two layer models. We first estimate the routing path of the prefix tokens and store their attention activations using KV-caching by sending our adversarial batch without the guessed token.

Then for each of the $2^{n \times L}$ possible expert allocations of the target token we query a local model with our target token and the previously computed KV-cache. Our local model supports disabling experts based on a given *routing path* $B$, a binary matrix of shape $n \times L$. We then store an expert allocation routing path table mapping each model output to its corresponding routing path bitmap, where each $b_{ij} \in B$ is equal to one if token $j$ is routed to expert $e_i$, and zero otherwise.

Then when querying the target model with our adversarial batch we compare the model output logits to our stored table and recover the token routing path. We expect to see a special distinguishable behavior in which the routing path of the *probe sequence* of the first adversarial batch query is not dropped. Instead of using exact matches in the routing path table, our approach looks for the nearest keys up to some $L_p$ distance to account for some of the floating point errors.

# E  CONDITIONAL ADVERSARIAL TOKEN-DROPPING

---

**Algorithm 1:** HIGH-LEVEL CONDITIONAL ADVERSARIAL TOKEN-DROPPING ALGORITHM

---

**Input:** Tokens Vocabulary $\mathcal{V}$, Number of experts $N$, Number of layers $L$, Capacity factor $C$,
$\qquad$ $M$ is the maximum sequence length, batch size $B$
**Output:** Secret user message $M$

1   $prefix \leftarrow$ "" // this is the prefix known to the attacker
2
3   $params \leftarrow (\mathcal{V}, N, L, C, M, B)$
    // making a guess about the next token
4   **for** *guess in $\mathcal{V}$* **do**
5      $probe\_sequence \leftarrow prefix + guess$
6      $min\_position \leftarrow get\_minimal\_position(message, params)$
      // check all of the possible experts
7      **for** *expert in experts* **do**
        // iterate over all possible positions for the token
8        **for** *position in [min_position, min_position + M]* **do**
          // collect all possible logits for different tokens dropped
9          routing_paths = *logits_to_routing_paths*(local_model, probe_sequence)
10
11         adv_batch $\leftarrow$ *construct_adv_batch*(local_model, probe_sequence, position, expert, params)
12         out1 = target_model (adv_batch, victim_message, probe_sequence_pos=0)
13         out2 = target_model (adv_batch, victim_message, probe_sequence_pos=1)
14
          // not dropped = 1, dropped = 0 <--> guess is correct
15         **if** *routing_paths[out1][expert] > routing_paths[out2][expert]* **then**
16           $prefix \leftarrow prefix + guess$
           // break out of positions and experts
17           **break**

---

In Algorithm 1, we give an algorithmic description of Conditional Adversarial Token-Dropping. This works as follows:

1. We guess the user input, token by token in order left to right;

2. We exploit **tie-handling** that happens **in the first MoE layer** of the `topk` implementation of the token to expert router;

3. Each expert has a limited buffer size of exactly $k$, these buffers are priority queues for the tokens, where the priority is assigned by the router;

4. Importantly, there exists a corner case where the router needs to decide how to deal with the tokens with exactly the same priority i.e. how to **handle ties**.

5. Internally this is not handled explicitly, but rather using a `topk` function e.g. `torch.topk` implemented in CUDA. Under the hood, `topk` uses a **"first comes first serves"** policy, in other words, for a pair of tokens with the same priority `topk` chooses the one that appears in the sequence earlier. We discuss this further in Appendix B.

6. This means, that **we can shape the expert buffer precisely to induce tie-handing**. From tie-handing we can infer the victims tokens by permuting the order of items in the batch i.e. checking model behaviour when the user input is appended before and after our *probe sequence*.

7. The two logits for the *probe sequence* will be different if we guessed the token correctly, and with that we can start attacking the next token.

The main idea is to target a specific expert, say $e_i$, then for a given guess for the next token, we could locally compute the priority $p_j$ of the guess appearing in this expert's buffer. With that we can look for sequences of tokens that are assigned to the same buffer with higher probability than $p_j$. These tokens will "block" our guess token, basically pushing its placement closer to the end of the buffer. By correctly guessing the placement of the token in the buffer excluding our blockers, we could precisely place the guess token on the edge of the buffer, and if the guess is correct it would compete with the real target token. This competition would yield different logits results as a function of the order of the inputs in the batch. We can then map the different logits to a routing path of expert assignment, and with that infer whether a dropout from the correct expert has occurred. We know shape the buffer in the first MoE layer and use the token-routing path-tracking to infer from the logits what has happened in this layer.

However, a number of things explicitly should be taken into account.

First, `topk` based tie handing relies on a `bitonic sort ()` that is unstable for short sequences and a stable sort for long (32+) sequences. Hence, the attacker batch should be comprised of three different inputs: (1) *probe sequence*: prefix + guess; (2) *blockers* that are used to block K positions on some target expert; finally, (3) *padding sequence* – some long sequence that forces the expert capacity to be $\geq 32$ to ensure sorting stability.

Second, careful management is required given that token placement needs to be precise. In paraticular, the adversary needs to take into account the expert capacity, the blockage from the prefix, blockage from the the blockers we get from the *probe sequence*. We launch the attack by the adversary searching for the blockers using a local copy of the model, recursively assuming some random user input.

Third, when dealing with more than one layer, we should take into account the possibility that the prefix tokens were dropped in previous layers, and thus even if we know the routing path of the guessed token, its' logits can change as a function of the prefix droppings in prior layers. Since we know the prefix, we can use our local model known upfront the prefix path. We need to send our adversarial batch to a local model, assume it is not drastically affected by the user input, and store the key-values cache for this execution. Then we send our guess tokens assuming different possible further expert token dropping combinations. With $L >> 3$-layers it seems completely infeasible – each layer out of $L$ layers has $n$ experts, and with Expert Choice Routing all $2^n$ assignments are possible. Thus each token $\in \mathcal{V}$ would have at most $2^{Ln}$ routing paths. By mapping our guessed tokens logits to the experts-path it went through, we can later observe the logits and infer if this logits comes from a routing path that include the target expert or not. We look for a change in inputs order, that would affect the guessed token routing path. We use a hooked local model, that allows us to disable experts given some token dropping routing path, we then bruteforce for all possible routing paths and store aside the logits to expert assignment mapping.

# F   Attack & Position performance

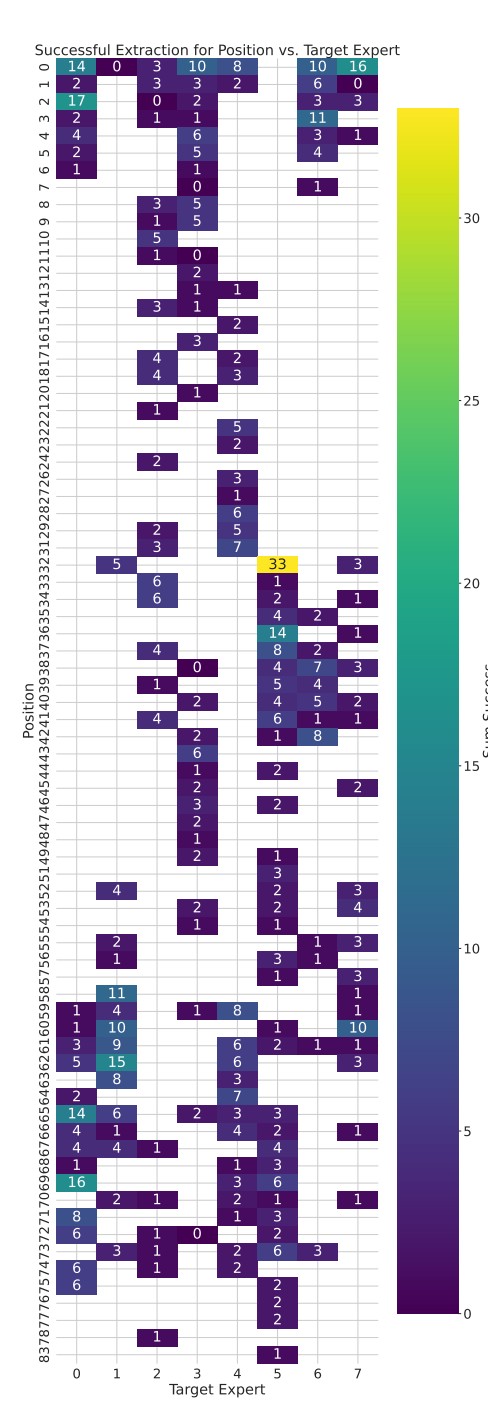

Figure 8: Heatmap showing the correlation between the expert and the position in the expert buffer for which the attack succeeds. Here, the attack progresses to the next token when any expert is successfully exploited to leak the token of the victim.

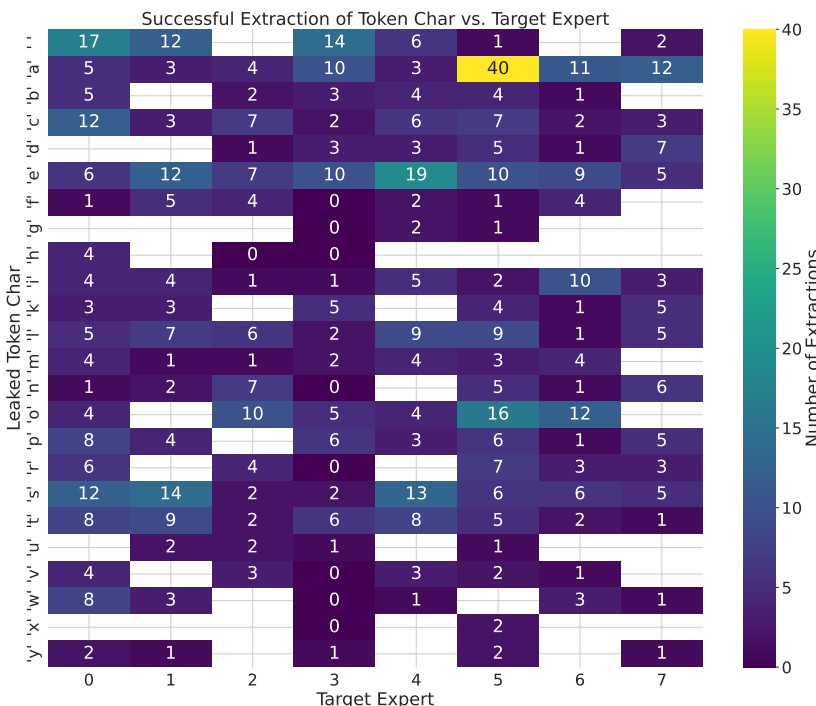

Figure 9: Heatmap showing the correlation between the expert and the token for which the attack succeeds. Here, the attack progresses to the next token when any expert is successfully exploited to leak the token of the victim.

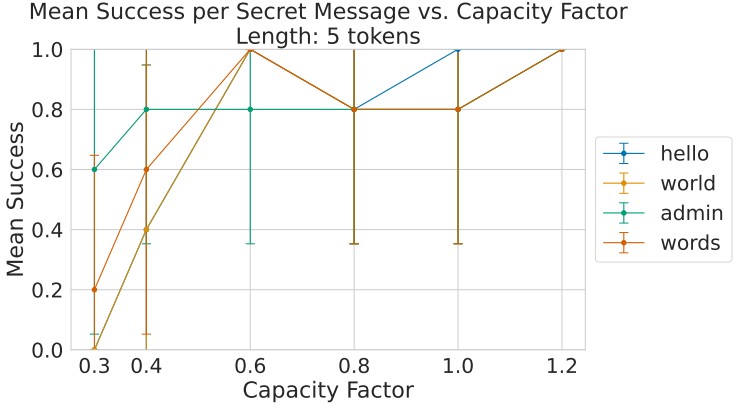

Figure 10: Individual message (length=5 tokens) performance with varying capacity factor.

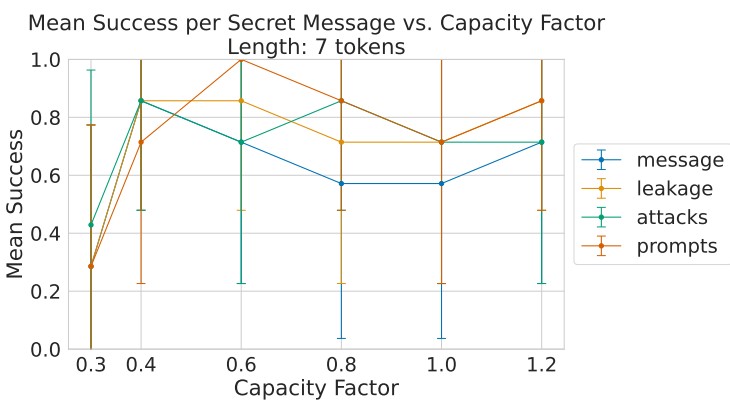

Figure 11: Individual message (length=7 tokens) performance with varying capacity factor.

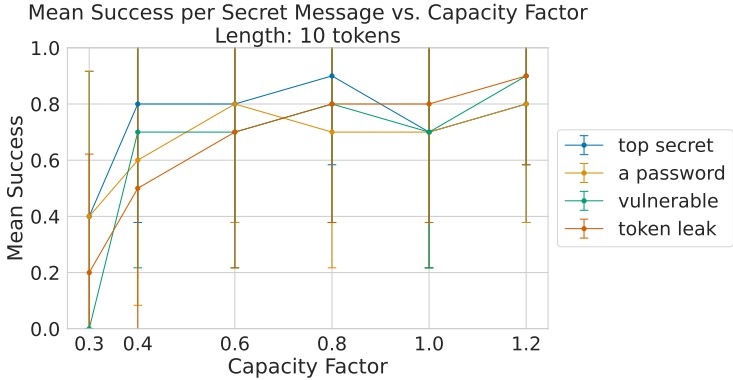

Figure 12: Individual message (length=10 tokens) performance with varying capacity factor.

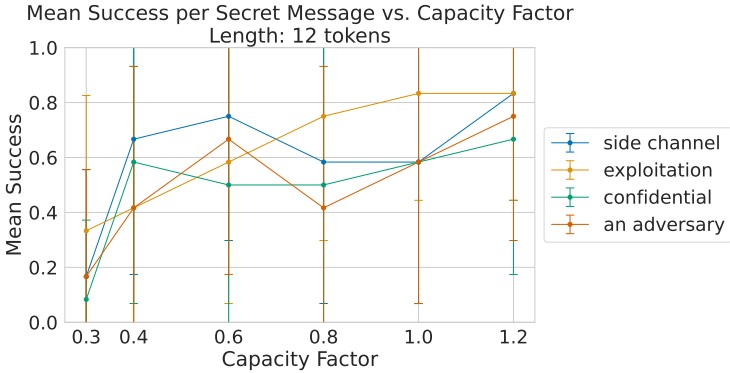

Figure 13: Individual message (length=12 tokens) performance with varying capacity factor.

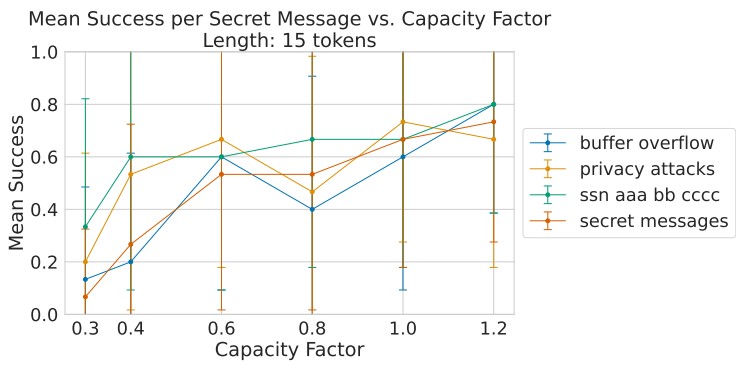

Figure 14: Individual message (length=15 tokens) performance with varying capacity factor.

