# OpenReview forum: "Stealing User Prompts from Mixture-of-Experts Models"
_ICLR.cc/2025/Conference — Submitted to ICLR 2025_

### Official Review · Reviewer_PLBR · 2024-10-28

**Soundness:** 3
**Presentation:** 4
**Contribution:** 3
**Rating:** 5
**Confidence:** 4

**Summary:**

In MoE models, individual experts process tokens in priority order; tokens with the same priority are processed in the arrival order (because of a CUDA quirk).  If the buffer is almost full, the second-to-arrive token is dropped.  This is a side channel: if an adversary can control the relative placement of their own and someone else's tokens in a batch, they can first fill the buffer with high-priority tokens, then switch the order between their own token and someone else's unknown token, and observe the resulting routings.  If the routing is the same for both tokens, this means the adversary's token is the same as the unknown token, revealing the value of the latter.  With repeated application, this can be leveraged into an extraction attack.

**Strengths:**

- Data-dependent computations are vulnerable to side-channel leakage: designers of ML systems need to learn this lesson.
- Cool exploitation of an interesting side channel in a particular MoE architecture (+ the top-k implementation in CUDA).
- History of computer security suggests that even seemingly impractical side channels can turn into exploitable vulnerabilities (with lots of additional research, of course).

**Weaknesses:**

- As acknowledged in the submission, the setting is unrealistic.  The adversary needs to (1) control the placement of target inputs in the batch, (2) repeatedly submit different orderings of the same batch to the model, and (3) observe its internal routing choices.   Man-in-the-middle (mention in 3.1) might be able to do (1) -- although not entirely clear how -- but not (2) or (3).  I cannot think of any setting where (2) and (3) are available to the adversary, yet the adversary is unable to directly observe inputs into the model.

- Evaluation is rudimentary, just a single Mixtral model.  I understand this is a proof-of-concept, but seems a little skimpy for a conference submission.

- Just a single routing strategy is investigated.  I do believe that other routing strategies may be similarly vulnerable, but again, seems skimpy for a conference submission.

- Defences are not really explored in any depth.  Randomizing top-k and/or token dropping (or other aspects) should mitigate the attack, but would it have a noticeable impact on performance / quality of the results?

**Questions:**

The paper seems premature in its current form, but I would advocate for it if a meaningful subset of the weaknesses were addressed.  It would require a much more substantial evaluation, though.

---

> ### Author Response · Authors · 2024-11-25
>
> Thank you very much for the feedback on the paper.
>
> > As acknowledged in the submission, the setting is unrealistic. The adversary needs to (1) control the placement of target inputs in the batch, (2) repeatedly submit different orderings of the same batch to the model, and (3) observe its internal routing choices. Man-in-the-middle (mention in 3.1) might be able to do (1) -- although not entirely clear how -- but not (2) or (3). I cannot think of any setting where (2) and (3) are available to the adversary, yet the adversary is unable to directly observe inputs into the model.
>
> We view our paper is an important proof-of-concept that model architectural decisions can have privacy impact on user submitted data. We argue in Section 6 that not only do we expect the attack to get better, but we also suspect that other flavours of MoE routing strategies can be exploitable.
>
> > Evaluation is rudimentary, just a single Mixtral model. I understand this is a proof-of-concept, but seems a little skimpy for a conference submission.
>
> Since our current attack algorithm is general (has precise complexity bound) and exploits specifics of the routing algorithm, we are not sure if further models would change the paper narrative at all. Importantly, complexity grows with the number of layers due to the logit matching phase of the attack and less with the choice of specific experts. We evaluated the attack with randomly initialised experts and saw no major change in hardness of blocking the experts.
>
> > Just a single routing strategy is investigated. I do believe that other routing strategies may be similarly vulnerable, but again, seems skimpy for a conference submission.
>
> We disagree with the reviewer on this point. Getting the a general attack algorithm to work and identifying the exact threat model to make it feasible was not at all a trivial effort that took many months of hacking and exploration. We also want to note that to make the attack work not only did we have to model the routing algorithm itself, but we also needed to debug deep specifics of how tie handling operates on accelerators in practice. We would expect that exploiting other strategies and optimising the one presented in our manuscript would take months if not years of effort; while at the same time the simple fact uncovered by our attack will remain -- MoE with ECR is vulnerable and leaks user data bit by bit.
>
> > Defences are not really explored in any depth. Randomizing top-k and/or token dropping (or other aspects) should mitigate the attack, but would it have a noticeable impact on performance / quality of the results?
>
> We want to note that adding random noise of magnitudes considered here will not disrupt the model performance almost at all, since at present the attack requires extreme precision to exploit the tie handling.

---

### Official Review · Reviewer_BQLL · 2024-11-01

**Soundness:** 2
**Presentation:** 3
**Contribution:** 2
**Rating:** 3
**Confidence:** 5

**Summary:**

The paper shows that if someone else's data is placed in the same batch as your data for many consecutive queries, and the model is a 2-layer MoE whose weights you have access to, and you can locally compute a forward pass on the MoE and the KV Cache, and that MoE is using cross-batch Expert-Choice Routing, and the router weights are heavily quantized in order to induce ties, and the MoE is running PyTorch TopK, then you can brute-force (with exponential query complexity) some of the tokens of the other person's query.

**Strengths:**

Attacking deployments of MoEs is a pretty interesting idea, and stealing the data of other users who are using the inference API is sufficiently high impact that this paper may have some impact even if the threat model and attack are unrealistic / impractical.

The diagrams explained the attack quite well.

**Weaknesses:**

The authors acknowledge upfront that their threat model is unrealistic (line 135).
I will add some additional reasons why the threat model is unrealistic;

- Not all deployed MoEs use Expert Choice Routing. In Expert Choice Routing, typically some tokens may be dropped if they don't go to any expert because that expert is filled. Expert Choice Routing can be very bad in some settings. The alternative is Dropless MoEs, which can be implemented in a couple different ways. I'm not sure which MoEs that are deployed actually use Expert Choice Routing, but if I were to go to an inference provider and ask for Deepseek MoE or DBRX, they would be serving a Dropless MoE. So some kind of table showing "here are the deployed MoEs that use Expert Choice Routing" would be useful. Of course this is closed information in many places, so I don't expect the authors to try and figure out whether Gemini or GPT-4 is using this, but you can at least go to all the inference providers serving open-weights MoEs (because you need open weights MoEs for this attack to work anyways) and see which ones use expert-choice routing. As far as I can tell, it is none of them, but I would want to see this table.
- Not all deployed MoEs would use the tie-handling mechanism that the attack relies on exploiting. The only way for a tie to occur is if two tokens have the exact same output from the router. But this does not happen even if those two tokens are actually the same, because over the course of an MoE with multiple layers, the token representations get mixed with other tokens via Attention. The authors note that they quantise the router weights to 5 bits to induce ties (line 377) but even if the router weights were quantised, you would not get ties in a multilayer model. I routed some tokens from Fineweb-CC-2014-03-04 through Mixtral 8x7B, saved the router scores, and found that there are basically no ties. If the authors could release their code that would be helpful to reproduce this tie-breaking behavior, even if it does require quantization.
- Some deployed MoEs would use jitter, which also totally messes up the proposed algorithm. Jitter just tries to sample from a slightly perturbed distribution so now we are even less likely to see ties.
- Not all deployed MoEs do not use the first-come-first-serve tie-breaking CUDA topk function that the authors assume they are using. For example, xAI's Grok and Gemini do not use this function. This is because the PyTorch TopK function on CUDA is absurdly memory inefficient. TRT, vLLM, etc. use other CUDA kernels for Topk that do not have this issue. Ex, NVIDIA's FasterTransformer uses this https://github.com/NVIDIA/FasterTransformer/blob/main/src/fastertransformer/kernels/sampling_topk_kernels.cu.
- Deployed MoEs typically do not have open weights. Even if we consider an inference provider running Pytorch on CUDA to serve an open-source MoE like Deepseekv2 such as Fireworks, the inference provider's KV Cache compression mechanism (anyone serving a model is not storing the full KV Cache, they are doing something like MLA, or sparse KV Cache, or quantized, or pruned, etc etc etc) is not publicly known. And this is required for the adversary to run this attack, because the adversary needs the KV Cache locally in the same way that the model is being inferenced on the cloud.
- If the adversary can run an open-weights MoE like Deepseek-v2 locally for many thousands of queries, they are operating with a massive amount of computational power. Furthermore, this attack needs the victim's data to also be present in the same batch for many queries.

The authors do not spend enough time proposing defenses; the paragraph starting on (line 484) should be expanded into a subsection. The authors had some ~30 lines remaining so it's not a matter of space constraints.

The main text of the paper is pretty much incomplete. There are too many places where the reader is forced to scroll to the Appendix and read a chunk of text in order to follow the paper. This is unfortunately becoming a common practice, but I dislike it nonetheless.

The confidence intervals seem way too large in Figure 4. It looks like all these attacks could just have 0 success rate. And this is even in the super unrealistic setting where the canaries are taking on a few values, the vocab is <10k (Gemma has vocab 256k), the model is artificially altered to make the attack work at all.

The attack is pretty unsophisticated. If I had to draw a comparison, I would say that this is like the brute-force binary search attacked proposed to extract logprobs by exploiting logit bias as proposed by Morris 2023. It's straightforward and if you don't care about efficiency it's fine, but it's not going to make an attack paper on its own. What can the community learn from the development from this attack? It has no practical implications, so there should be something about the design that is clever or inspires new ideas.

There are some minor typos (line 496) (line 837) (line 342) (line 819) (line 820)

**Questions:**

n/a

---

> ### Author Response · Authors · 2024-11-25
>
> Thank you very much for the feedback on the paper.
>
> > The authors acknowledge upfront that their threat model is unrealistic (line 135). I will add some additional reasons why the threat model is unrealistic; ...
>
> Thank you very much for all of the above points, they are extremely useful for setting the scene on deployment practices. We agree fully that at present the attack is unrealistic and we emphasize this in the paper. We will add a section to the appendix to expand on the above points and also add a note to the existing Section 6 with additional requirements for the future attacks to become realistic. We once again thank the reviewer for this.
>
> > The authors do not spend enough time proposing defenses; the paragraph starting on (line 484) should be expanded into a subsection. The authors had some ~30 lines remaining so it's not a matter of space constraints.
>
> Many thanks for this. We will add more to the defenses section. Currently to defend one really needs to add minor amount of noise, since the current attack requires extreme precision to work and errors compound making extraction of later letters harder.
>
> > The main text of the paper is pretty much incomplete. There are too many places where the reader is forced to scroll to the Appendix and read a chunk of text in order to follow the paper. This is unfortunately becoming a common practice, but I dislike it nonetheless.
>
> We agree with the reviewer, but we could not find another way to overcome it, since there is quite a bit of complexity to describe how and why the attack works. We will iterate over the manuscript to improve its readability.
>
> > The confidence intervals seem way too large in Figure 4. It looks like all these attacks could just have 0 success rate. And this is even in the super unrealistic setting where the canaries are taking on a few values, the vocab is <10k (Gemma has vocab 256k), the model is artificially altered to make the attack work at all.
>
> Indeed. That was due to the quirks of the hyperparameters chosen for this particular evaluation run. After a minor change to the evaluation parameters we reduced the variance in performance and now the attack works in almost 100% of cases.
>
> > The attack is pretty unsophisticated.
>
> Although we agree with the reviewer that the final attack algorithm is unsophisticated, we want to stress that making it work was not at all easy. Getting the general attack algorithm to work and identifying the exact threat model to make it feasible was not at all a trivial effort that took many months of hacking and exploration. We also want to note that to make the attack work not only did we have to model the routing algorithm itself, but we also needed to debug deep specifics of how tie handling operates on accelerators in practice.
>
> > What can the community learn from the development from this attack? It has no practical implications, so there should be something about the design that is clever or inspires new ideas.
>
> We view our paper is an important proof-of-concept that model architectural decisions can have privacy impact on user submitted data. We argue in Section 6 that not only do we expect the attack to get better, but we also suspect that other flavours of MoE routing strategies can be exploitable. Our work is an example to a simple fact -- *MoE with ECR is vulnerable and leaks user data bit by bit*.
>
> Note that you could make the same comments about Spectre or Meltdown attacks (ie simple, unsophisticated, leaks a single bit), yet it had a major impact in how speculation is performed on our everyday computers; and it was in fact used to perform real-world impacting attacks. Our work shows a novel kind of vulnerability; stronger attacks will follow.
>
> > There are some minor typos (line 496) (line 837) (line 342) (line 819) (line 820)
> Thank you very much for this. All fixed now.

---

### Official Review · Reviewer_B26R · 2024-11-04

**Soundness:** 2
**Presentation:** 3
**Contribution:** 3
**Rating:** 5
**Confidence:** 3

**Summary:**

This paper explores a novel security vulnerability in Mixture-of-Experts (MoE) language models, specifically focusing on the risk of prompt leakage through the architecture's routing mechanisms.The proposed attack, an adversary manipulates expert buffers within an MoE model to extract a victim's prompt by observing how token routing and dropping affect model outputs. The study reveals that an attacker can reconstruct a user’s prompt by exploiting token-dropping patterns and guessing tokens sequentially.

**Strengths:**

- The study introduces a novel security concern by identifying a previously unexamined vulnerability in LLM service.
- Experimental results demonstrate the effectiveness of the proposed attack, showing that it reliably extracts user prompts under the specified conditions.

**Weaknesses:**

- The threat model assumes an attacker with significant control over the LLM server, which may not be practical in real-world settings. Additionally, token-dropping techniques are not widely used in recent LLM inference architectures, limiting the relevance of the attack to current models.
- The attack is computationally intensive, requiring up to 1,000 tokens for each token being extracted, which may restrict its feasibility in large-scale applications.
- The explanation of the proposed method for Recovering Target Token Routing Path lacks clarity. It is unclear how the method handles cases where two tokens share the same routing path. If two tokens follow identical paths, this could complicate the attack, as distinguishing between them based on routing alone may not be difficult.

**Questions:**

- Could you please further discuss about how man-in-the-middle attacks can help to inject the proposed attack in LLM server?
- Could you discuss what will happen if there are two tokens sharing the same routing path.

---

> ### Author Response · Authors · 2024-11-27
>
> Thank you very much for the feedback on the paper.
>
> > Could you please further discuss about how man-in-the-middle attacks can help to inject the proposed attack in LLM server?
>
> Thank you for raising this point. We agree that discussing the feasibility of the attack in real-world scenarios is important. In MITM scenarios an attacker can have more control of the user interaction with a server [1], even though the communication is encrypted and thus not visible to the attacker. In such settings one can carry our attack more realistically, as the requirement of controlling the positions in batch or forcing the user to send the same secret message repeatedly. We will add a section to illustrate that.
>
> > Could you discuss what will happen if there are two tokens sharing the same routing path.
>
> Thank you for this insightful question.  Our exploit is designed to handle cases where two tokens share the same routing path. We ensure a unique signal is generated only when our guesses for both the token's identity and its position within the target expert buffer are correct. This allows us to differentiate between tokens even if they follow identical routing paths. We will clarify this mechanism further in the revised version.
>
> > The threat model assumes an attacker with significant control over the LLM server, which may not be practical in real-world settings. Additionally, token-dropping techniques are not widely used in recent LLM inference architectures, limiting the relevance of the attack to current models.
>
> We appreciate your feedback on the threat model.  You're right that the current attack assumes a strong attacker, and token-dropping may not be prevalent in current LLMs. However, we believe it's crucial to proactively identify and address potential vulnerabilities, even if they are not immediately exploitable.  Our work aims to raise awareness about the risks associated with certain design choices in LLMs and encourage the development of secure and robust architectures.  This is particularly relevant for future LLMs and evolving inference techniques, where token-dropping or similar mechanisms might be employed. By highlighting these vulnerabilities, we hope to inform the design and implementation of secure LLMs, even if the specific attack demonstrated here has limitations in current real-world settings.
>
> > The attack is computationally intensive, requiring up to 1,000 tokens for each token being extracted, which may restrict its feasibility in large-scale applications.
>
> We acknowledge the computational limitations of the current attack. As you pointed out, the attack's complexity could hinder its feasibility in large-scale applications. However, we believe the core findings of our research remain valuable.  Our work highlights a previously unknown vulnerability that could have significant implications for the security and privacy of LLMs.  This information is crucial for LLM architects and developers, particularly those working with Trusted Execution Environments, as it underscores the need for robust security measures to protect user data, even within batched processing environments. We hope our findings will stimulate further research into more efficient attack strategies and mitigation techniques.
>
> > The explanation of the proposed method for Recovering Target Token Routing Path lacks clarity. It is unclear how the method handles cases where two tokens share the same routing path. If two tokens follow identical paths, this could complicate the attack, as distinguishing between them based on routing alone may not be difficult.
>
> Thank you for your feedback on the clarity of our method. We will revise the explanation of the 'Recovering Target Token Routing Path' method in the paper to provide a more comprehensive and clear description. As mentioned earlier, our attack can successfully distinguish between tokens sharing the same routing path by relying on the unique signal generated only when both the token's identity and position are correctly guessed. We will ensure this aspect is clearly conveyed in the revised manuscript.
>
> [1] https://docs.google.com/presentation/d/11eBmGiHbYcHR9gL5nDyZChu_-lCa2GizeuOfaLU2HOU/edit#slide=id.g1d134dff_1_222

---

> > ### Comment · Reviewer_B26R · 2024-11-30
> >
> > Thank you for your further explanation. I agree that identifying potential risks in LLM systems is valuable work. However, the impact of this approach seems limited, as it is tailored to a less commonly used method—token drop—and is not very practical due to its computational expense. The application of strong attacker settings further contributes to these limitations.
> >
> > I decide to keep my score at 5.

---

### Meta-Review · Area_Chair_KiWh · 2024-12-23

**Metareview:**

This paper presents a new prompt stealing attack for LLMs with MoE architecture. The attack exploits the expert-choice routing mechanism to disclose a victim user's prompt when the attacker's input is batched together with the victim's. This paper is the first to show such a side-channel attack is possible.

All reviewers agree the attack presented in this paper is novel and interesting, and can serve as a cornerstone for future attacks that exploit similar side-channel vulnerabilities. However, reviewers also pointed to several major weaknesses, including limited generality of the attack, excessive query cost, and evaluation lacking depth. AC agrees with the paper's merits and shortcomings and believes the paper has limited impact in its current form. If these weaknesses are addressed, especially if the attack can be expanded to handle more general MoE architectures with reduced query cost, it can be a much more influential paper.

**Additional Comments On Reviewer Discussion:**

Reviewers and authors discussed the shortcomings, and the authors admitted the above limitations. The paper's decision ultimately depended on whether reviewers believed the techniques presented in the paper have future impact even if they may be unrealistic in their current form.

---

### Decision · Program_Chairs · 2025-01-22

Reject